# Anti-Idiotype scFv Localizes an Autoepitope in the Globular Domain of C1q

**DOI:** 10.3390/ijms22158288

**Published:** 2021-08-01

**Authors:** Nadezhda Todorova, Miroslav Rangelov, Vanya Bogoeva, Vishnya Stoyanova, Anna Yordanova, Ginka Nikolova, Hristo Georgiev, Daniela Dimitrova, Safa Mohedin, Katerina Stoyanova, Ivanka Tsacheva

**Affiliations:** 1Institute of Biodiversity and Ecosystem Research (IBER), Bulgarian Academy of Sciences, Yurii Gagarin Str. Bl. 2, 1113 Sofia, Bulgaria; nadeshda@abv.bg; 2Institute of Organic Chemistry with Centre of Phytochemistry (IOCCP), Bulgarian Academy of Sciences, Acad. G. Bonchev Str. Bl. 9, 1113 Sofia, Bulgaria; marangelov@gmail.com; 3Institute of Molecular Biology, Bulgarian Academy of Sciences, Acad. G. Bonchev Str. Bl. 21, 1113 Sofia, Bulgaria; vanya.bogoeva@gmail.com (V.B.); anna_gurova@bio21.bas.bg (A.Y.); 4Department of Chemistry, Biochemistry, Physiology and Pathophysiology, Medical Faculty of Sofia University, 1 Kozyak Str., 1407 Sofia, Bulgaria; vishnya_stoyanova@abv.bg; 5Department of Biochemistry, Faculty of Biology, Sofia University, 8 D. Tsankov Str., 1164 Sofia, Bulgaria; ginka.nikolova@biofac.uni-sofia.bg (G.N.); sofibg@hotmail.com (S.M.); katerina_stoyanova_@abv.bg (K.S.); 6Institute of Immunology, Hannover Medical School, Carl-Neuberg Str. Bl. 1, 30625 Hannover, Germany; georgiev.hristo@mh-hannover.de; 7Institute of Biophysics and Biomedical Engineering, Department of Electroinduced and Adhesive Properties, Bulgarian Academy of Sciences, Acad. G. Bonchev Street, Bl. 21, 1113 Sofia, Bulgaria; daniadim@yahoo.com

**Keywords:** C1q, globular autoepitopes, scFv, anti-DNA

## Abstract

We addressed the issue of C1q autoantigenicity by studying the structural features of the autoepitopes recognized by the polyclonal anti-C1q antibodies present in Lupus Nephritis (LN) sera. We used six fractions of anti-C1q as antigens and selected anti-idiotypic scFv antibodies from the phage library “Griffin.1”. The monoclonal scFv A1 was the most potent inhibitor of the recognition of C1q and its fragments ghA, ghB and ghC, comprising the globular domain gC1q, by the lupus autoantibodies. It was sequenced and in silico folded by molecular dynamics into a 3D structure. The generated 3D model of A1 elucidated CDR similarity to the apical region of gC1q, thus mapping indirectly for the first time a globular autoepitope of C1q. The V_H_ CDR2 of A1 mimicked the ghA sequence GSEAD suggested as a cross-epitope between anti-DNA and anti-C1q antibodies. Other potential inhibitors of the recognition of C1q by the LN autoantibodies among the selected recombinant antibodies were the monoclonal scFv F6, F9 and A12.

## 1. Introduction

C1q, the recognition molecule of the classical complement pathway, appears as an autoantigen in several human autoimmune disorders, most notably in systemic lupus erythematosus (SLE) [1,2,3,4]. SLE is characterized by a broad spectrum of autoantibody specificities, some of which are directed against nuclear components and C1q. Anti-dsDNA autoantibodies, the hallmark of SLE, are hypothesized as a result of defective removal of apoptotic material, eventually resulting in an immune response to these normally sequestered autoantigens [5]. The anti-C1q autoantibodies, which closely follow the appearance of anti-dsDNA, are hypothesized as a result of conformational changes in C1q due to immobilization and exposure of neo-epitopes [6,7,8], underlain by increased hydrophobicity [9] or/and as a result of post-translational modifications [10].

Anti-C1q autoantibodies are associated with lupus nephritis (LN), a clinical condition in more than 30% of SLE patients [11]. LN follows a course of exacerbations and remissions and more than 25% of LN patients experience multiple episodes of active nephritis at increased risk of progressing to end stage renal disease [12].

The anti-C1q autoantibodies are known to be polyclonal high affinity IgG molecules, produced in an antigen-driven process [13,14,15]. The complex structure of C1q provides two types of functional domains—the N-terminal collagen-like region (CLR) and the C-terminal globular heads ghA, ghB and ghC, associated in a globular domain (gC1q). Autoepitopes for the anti-C1q antibodies have been detected on both of them [15,16,17,18]; however, so far only a linear epitope in CLR has been localized [19]. The localization of epitopes in gC1q is challenging due to the fact that ghA, ghB and ghC are structurally and functionally independent modules [20]. The heterotrimeric organization of gC1q offers functional flexibility and possible versatility of epitopes for recognition by the autoantibodies. The autoantigenic behavior of C1q is reflected by specific features of its 3D structure. Accordingly, the localization of C1q autoepitopes and studying their structural features would largely contribute to our understanding of C1q autoantigenicity and would suggest the molecular events leading to the generation of autoantibodies to C1q.

In the present study, we addressed the issue of C1q autoantigenicity by studying the structural features of C1q autoepitopes that are recognized by human polyclonal anti-C1q autoantibodies present in the sera of LN patients. In our previous research, we have found that LN sera were positive for anti-C1q antibodies that recognized different combinations of C1q domains and their smaller fragments, e.g., CLR and the globular fragments of A, B and C chains comprising gC1q—ghA, ghB and ghC, along with the intact molecule. Presumably, these sera contained anti-C1q antibodies with different epitope specificities [15]. Consequently, we combined the sera with similar epitope specificities and used the subgroups of anti-C1q antibodies, isolated from them, to generate anti-idiotype scFv antibodies which we applied as tools to localize C1q autoepitopes. For some scFv antibodies, we detected inhibitory activity on the recognition of immobilized C1q by human LN autoantibodies using immunosorbent and spectrofluorometric analyses. By molecular dynamics simulation we generated a 3D model of the most potent inhibitor—the monoclonal scFv A1, which mapped to the apical region of gC1q comprising parts of ghA, ghB and ghC.

## 2. Results

### 2.1. Generation of Anti-Idyotipic scFv

Based on our previous study of the epitope specificity of the polyclonal autoantibodies to C1q from LN sera [15], we classified the analysed sera into six groups according to their positivity for the used test-antigens: intact C1q, CLR and ghA, ghB and ghC as representatives of gC1q. Sera that were positive for the intact C1q (pooled in group A) became the source of anti-C1q designated fraction A (Table 1). Sera positive for ghC (pooled in group D) yielded autoantibodies designated fraction D. Sera that were positive for combinations of test-antigens were pooled in other groups as follows: group B (C1q + CLR), group C (C1q + CLR + ghC), group E (CLR + ghA/ghB + ghC) and group F (ghA/ghB + ghC), and became sources of anti-C1q fractions B, C, E and F, respectively. Antibodies from each group of pooled sera were affinity purified and used as antigens for the selection of monoclonal scFv, part of which we expected to be anti-idiotypic and therefore potential structural analogues of C1q autoepitopes. The anti-C1q fractions A, B and C were extracted by incubation of respective pooled sera diluted in PBS containing 0.75M NaCl with immobilized intact C1q, as it was the common test-antigen for these groups, while anti-C1q fractions D, E and F were extracted likewise on immobilized ghC, the common test-antigen for the rest sera groups. High ionic strength was needed to inhibit the IgG molecules that tended to interact with C1q via Fc and to allow the interaction of IgG molecules via their Fab, thus ensuring their anti-C1q specificity. The affinity procedure was performed before every round of selection and the presence of anti-C1q IgG antibodies was routinely checked by immunoblotting against the whole set of test-antigens (data not shown). Interestingly, eluted anti-C1q fractions (A–C) gave a strong signal not only to intact C1q but also to ghA, while eluted anti-ghC fractions (D–F) gave comparable signals in their binding to intact C1q, ghA, ghB and ghC. After three rounds of selection of Griffin.1 library [21], we rescued 68 monoclonal scFv antibodies and tested them for antigen binding—10 clones selected for anti-C1q fraction A (A1 to A10), 13 clones selected for fraction B (B1 to B12 and A12), 12 clones selected for fraction C (C1 to C12), 11 clones selected for fraction D (D1 to D11), 11 clones selected for fraction E (E1 to E11) and 11 clones selected for fraction F (F1 to F11). Twenty-one of them were specific for their respective antigens (Table 2). Phage particles from each specific scFv clone were transferred to non-suppressor strain *E.coli* HB2151 and expressed in soluble form. The antigen specificity of the soluble scFv was confirmed by the detection of both scFv tags—C myc and His (Table 3). We ruled out the scFv clones which were specific for isotypic epitopes by using a negative control of pooled human IgG from healthy donors, thus selecting 14 anti-idiotypic scFv clones.

### 2.2. ELISA Analysis for Selection of Inhibitory scFv

We analysed the selected anti-idiotypic clones as potential inhibitors on the recognition of immobilized C1q by the LN autoantibodies. The large-scale analysis was done with partially purified scFv by ammonium sulfate precipitation. The source of autoantibodies was affinity purified IgG from a pool of all analysed LN sera, designated IgG_LN_. The analysis was performed by competitive ELISA, with increasing amounts of the competitors. We found four clones with inhibitory capacity: A1 (anti-fraction A), A12 (anti-fraction B), F6 and F9 (anti-fraction F) (Figure 1, Appendix A). Clone A1 showed the greatest extent of inhibition by the smallest amount of introduced competitor. Consequently, it was expressed at a large scale and purified by affinity Ni+ chromatography. The purified scFv A1 exhibited 47% inhibition on the recognition of immobilized C1q by the LN autoantibodies (Figure 2A). Furthermore, A1 inhibited to a greater extent the recognition of ghA, ghB and ghC by the same autoantibodies (Figure 2B). This data suggested that A1 was structurally analogous to an autoepitope located in gC1q. The inhibition of all three globular heads by A1 indicated that they all contributed to the formation of a globular autoepitope, thus suggesting its conformational nature.

### 2.3. Fluorescence Spectroscopy Analysis

The inhibitory capacity of scFv A1 was verified by fluorescent spectroscopy. LN autoantibodies complexed with scFv A1 inhibited the binding of C1q, ghA, ghB and ghC, respectively, when applied in increasing concentrations.

The inhibitory capacity of scFv A1 was clearly detected by its ability to block the recognition of C1q by LN autoantibodies (Figure 3). Extrinsic fluorescence of the hydrophobic dye ANS was used for the experimental study in order to avoid the intrinsic protein fluorescence. In Figure 3A, the ANS spectrum is shown (1, lowest spectrum), LN autoantibodies complexed with scFv A1 (2, upper spectrum) and C1q (3–6) in increasing concentrations (0.03 µM–0.3 µM). The blocking effect is shown by C1q spectra that have overlapped the spectrum of the complexed autoantibodies with scFv A1.

Similar to C1q binding, we performed experiments with ghA, ghB and ghC fragments. The results showed inhibitory effects of the studied proteins (Figure 3B,C). Experimental data with ghB and ghC fragments had similar results, showing inhibition of scFv A1 upon the ghB and ghC.

### 2.4. Sequence Analysis of the Selected scFv A1

The scFv A1 sequence, containing only antibody coding parts, was further analysed. EMBOSS sequence tools revealed only 1 ORF, standard codon table, stretched on the whole length of the scFv (223 amino acids).

The sequence was compared to the human immunoglobulin germ-line V-gene sequences and the CDR prediction analysis was performed with VBASE2 platform [22]. Heavy chain V-segment are of the IGHV4 family, while the light chain is classified as IGKV1 subgroup. No J-segment is discovered in the V_L_; however, the junction in V_H_ participated as expected in the CDR3 formation.

### 2.5. ScFv A1 Homology Model

The sequence analysis of scFv A1 further continued with generating its homology model (Figure 4) in order to detect possible molecular similarity with the C1q molecule. Structural comparison of C1q globular heads and scFv A1 was performed on V_H_, as V_H_ CDR3 was expected to exhibit the greatest degree of variety and, accordingly, to have a distinctive role in antigen recognition. The comparison was based on the assumption that the reacting amino acids were situated in flexible loops of both structures, rather than the structurally compact and rigid beta-sheet structures. CDR1, 2 and 3 were compared loop by loop with C1q separate chains loops. The similarity is based on the polarity of the amino acids (Figure 5). The backbone is analysed only as amino acid side chain (R) polarity due to the mobility of R in solution, and not as a chain position in the loop. A snapshot of the described procedure is depicted in Figure 6.

All CDR loops have their most similar residues in regions involved in the C1q interfaces, namely between C- and D-, G-/H- and E-/F- loops according to Gaboriaud et al. [23]. Notably, these parts are not conserved in C1q and similar proteins, but rather a substantial variability in the residues is found at the interfaces, thus eliminating occasional likeness. Based on the loop similarity model, all of the three CDR regions of scFv VH have analogs in the apex of native gC1q chains (Figure 7) that are known to exhibit differences in their surface patterns, with respect to both charged and hydrophobic residues. These findings indicate the sites on the exposed surface of gC1q, which are important for the engagement of lupus antibodies and either contribute to or lie in close proximity with detected C1q parts, similar to scFv A1 CDRs. The mimicry of scFv A1 to parts of all globular heads ghA, ghB and ghC reveals possible involvement of all of them in the interaction with the lupus autoantibodies and is consistent with the inhibitory activity of scFv A1 towards all three globular heads, thus confirming the conformational nature of the mimicked autoepitope.

## 3. Discussion

We generated scFv A1 as anti-idiotypic antibody to a fraction of LN anti-C1q autoantibodies and it turned out to be a potent inhibitor of the recognition of C1q, ghA, ghB and ghC by the LN autoantibodies. The generated 3D model of A1 sequence elucidated CDR similarity to the apical region of gC1q, thus mapping indirectly for the first time a globular autoepitope of C1q. The similar regions are situated in the three globular heads, suggesting that the globular autoepitope may consist of a number of linear stretches, thus being a conformational one. This is consistent with the fact that the anti-C1q, named fraction A, were extracted from LN sera, positive only for the intact C1q. Presumably, each of the globular heads is of low affinity when binding the LN autoantibodies and, consequently, none of them was detected as individual test-antigen.

Our experimental data suggest also that there are other potential inhibitors of the recognition of C1q by the LN autoantibodies among the selected recombinant antibodies, namely the monoclonal scFv F6, F9 and A12 (Figure 1). The clones, F6 and F9, are likely to mimic an autoepitope within gC1q given that the anti-C1q fraction F (Table 1) contains autoantibodies specific for ghA and/or ghB and ghC. The clone A12 is anti-idiotypic to anti-C1q fraction B containing autoantibodies specific for C1q and CLR, consequently it is expected to mimic an epitope within CLR.

Anti-C1q autoantibodies have been researched for a long time [25]. The focus of the research for the best part of that time was the CLR of C1q because the anti-CLR autoantibodies strongly correlated with disease activity [26,27,28,29,30,31]. However, the detection of autoantibodies binding gC1q opened up new possibilities for an insight into C1q autoantigenicity. Anti-gC1q antibodies, unlike anti-CLR, were found to be prevalent in non-active LN patients [15]. Moreover, again unlike anti-CLR, we detected these autoantibodies in high titers among healthy humans [32]. Collectively, these data brought us to the assumption that the appearance of anti-gC1q antibodies would act more as a trigger rather than a mediator of the clinical outbreak of C1q-associated autoimmunity. As C1q is an extracellular antigen, its autoantigenicity does not fit in the concept of the resonance hypothesis [33], according to which the tissue targeting generates additional autoantigens that fuel further autoimmune response. It has been reported that autoantibodies develop before the clinical onset [34,35] and among the first specificities to appear are anti-dsDNA antibodies which also exhibit the highest sensitivity for diagnosing SLE. The loop similarity of scFv A1 and gC1q revealed that V_H_ CDR2 of scFv A1 mimics the ghA sequence GSEAD, which was suggested as a cross-epitope between anti-DNA and anti-C1q antibodies [24]. We hypothesized that the cross-reacting anti-dsDNA could be the antibodies dragging C1q initially into the context of autoimmune response by turning it into an autoantigen. This assumption agrees with the observations that, chronologically, anti-C1q autoantibodies appear after anti-dsDNA autoantibodies [34,35]. It is also likely that the binding of these autoantibodies with gC1q would introduce the first conformational changes that could lead to a subsequent exposure of neo-epitope in distant parts of the molecule.

Previously, we have found that conformational disturbance within ghB affected the degree of autoantigenicity of the whole C1q molecule [9]. If our hypothesis is supported by further analysis, it would mean that anti-gC1q autoantibodies are the first to appear in the autoimmune setting involving C1q as autoantigen. This type of initiating factor is unlikely to act alone but rather in combination with others. Recently, Csorba et al. reported that anti-C1q antibodies specific for the major linear epitope of CLR are cross-reacting with EBNA-1 (Epstein–Barr virus nuclear antigen 1)-derived peptide, implying that anti-C1q in SLE could be induced through molecular mimicry by EBV [36].

Interestingly, the BLAST analysis of the scFv A1 sequence revealed 91% similarity in the CDR regions of DQ201310.1, H3 Ab, which is an anti-EBV antibody [37]. It recognizes one of the EBV proteins that resembles the tumor necrosis factor (TNF) receptor and is essential for virus-induced cell signaling pathways. Thus, we can hypothesize that gC1q, in addition to CLR, is also involved in the EBV-induced autoimmune microenvironment.

The waste-disposal hypothesis [38] appointed the clearance of apoptotic cells as a major event in triggering autoimmunity. C1q participates in that clearance by binding apoptotic cells directly via its gC1q [39,40]. Indirectly, C1q can also clear apoptotic cells bound by IgM [41,42,43,44]. The latter is a natural ligand of C1q for the activation of the classical complement pathway and the IgM-binding site is also located in gC1q [45]. Consequently, deficiency of C1q results in impairment of apoptotic cell clearance which provides autoantigenic stimuli for autoreactive B cells with diverse specificities [46,47]. The presence of anti-C1q antibodies could cause a functional deficiency of C1q. Anti-C1q autoantibodies from LN patients were reported to inhibit the clearance of apoptotic cells in vitro [48]. Given the apoptotic blebs are recognized by gC1q, it seems likely that the inhibiting autoantibodies are anti-gC1q. The revealed structural similarity of scFv A1 with the apical region of gC1q opens up a possibility to test the hypothesis that anti-gC1q interferes with the mechanism of apoptotic cell clearance by C1q, both directly and indirectly, due to overlapping of the globular autoepitope with the apoptotic cell-binding site and/or IgM-binding site of C1q.

## 4. Materials and Methods

### 4.1. Buffers

The following buffers were used: sodium carbonate buffer (SC), pH 9.6; PBS containing 0.1% Tween 20 (TPBS); PBS with 0.75 M NaCl (PBS/0.75); elution buffer (EB) [0.1 M Glycine-HCl, pH 2.8]. 1-Anilinonaphthalene-8-sulfonic acid (ANS) was purchased from Sigma (St. Louis, MO, USA) and its concentration was determined spectrophotometrically by its extinction coefficient at λ_370_ nm (ε_370_ = 6.8 × 10^3^ M^−1^ cm^−1^).

### 4.2. Growth Media

The following growth media were used: 2xTY medium (20 g/L Tryptone (Casitose Type-I, HiMedia, Mumbai, India), 10 g/L Yeast Extract (HiMedia, Mumbai, India), 20 g/L NaCl (Fisher Scientific, Loughborough, UK), pH 7.2) containing 100 μg/mL Ampicillin Ampicillin (Fisher Bioreagents, Pittsburg, PA, USA) and supplemented with 1 mM MgSO_4_ (Fisher Scientific, Loughborough, UK) and 1% Glucose (Merck KGaA, Gernsheim, Germany) (2xTY-Amp); LB medium (10 g/L Tryptone, 5 g/L Yeast Extract, 10 g/L NaCl, pH 7.2) containing 100 μg/mL Ampicillin and supplemented with 1 mM MgSO_4_ and 1% Glucose (LB-Amp); ZYP-5052 medium for autoinduction (1% Tryptone, 0.5% Yeast Extract, 25 mM (NH_4_)_2_SO_4_ (Chem-Lab NV, Zedelgem, Belgium_)_, 50 mM KH_2_PO_4_ (Fisher Bioreagents, UK_)_, 50 mM Na_2_HPO_4_ (Fisher Bioreagents, Loughborough, UK), 0.5% glycerol (Sigma, St. Louis, MO, USA), 0.05% glucose, 0.2% α-lactose (Carl Roth GmbH, Karlsruhe, Germany), 1 mM MgSO_4_), containing 100 μg/mL Ampicillin; TYE agar (10 g/l tryptone, 5 g/l yeast, 8 g/l NaCl, 15 g/l agar), containing100 μg/mL Ampicillin (TYE-Amp).

### 4.3. Expression of the Globular Heads ghA, ghB and ghC Representing gC1q

The recombinant C1q globular heads ghA, ghB and ghC were expressed as fusion proteins with bacterial MBP in *E. coli* BL21 (DE3) and purified, as described previously [17].

### 4.4. Preparation of Antigens for the Selection of Anti-Idiotypic scFv

Six fractions of anti-C1q antibodies were affinity isolated from previously analyzed LN sera [15] according to their epitope specificity (Table 1). C1q-coated (4 μg/well) plates, blocked with 1% BSA, were incubated overnight at 4 °C with 250 μL/well of pooled sera from groups A, B and C, diluted 1:50 in PBS/0.75. The wells were washed three times with TPBS and once with PBS. Bound anti-C1q was eluted by incubation of each well with 50 μL/well EB for precisely five minutes and was immediately neutralized. Forty-eight C1q-coated wells were used for each group of pooled sera and the eluted 2.4 mL for every type of anti-C1q fractions were neutralized with 90 μL of 1.5 M Tris (Fisher Bioreagents, NJ, USA) (pH 8.8) and dialysed against PBS.

Analogously, the pooled sera from groups D, E and F were incubated each on 48 ghC-coated wells (4 µg/well), blocked with 1% BSA. This affinity isolation resulted in six anti-C1q fractions which were used as antigens for the selection of anti-idiotypic scFv antibodies.

### 4.5. Screening of Human scFv Phagemid Library Griffin.1

Panning of phage scFv antibodies was performed following [21]. Briefly, immunotubes (NUNC MaxiSorp) were coated overnight with 2.4 mL of each of the six antigen groups of anti-C1q preparations. The tubes were blocked with 0.5% Tween 20 (Fisher Bioreagents, Fair Lawn, NJ, USA) for two hours and incubated with 1 mL (approximately 10^13^ PFU) of phage scFv in 0.5% Tween 20 for 30 min with rotation, followed by a stationary incubation for an additional 90 min. All incubation steps were performed at room temperature. The tubes were washed 10 times with TPBS and 10 times with PBS. Bound phage scFv were eluted with 1 mL of 100 mM triethylamine and gentle rotating of the tube for 10 min. The eluted material was neutralized with 0.5 mL 1 M Tris-HCl (pH 7.4). Half of the eluted phage suspension was used to infect 10 mL log phase *E. coli* TG1 cells at 37 °C for 30 min, and then plated on TYE-Amp plates for an overnight cultivation. The infected cells were transferred to 100 mL 2xTY-Amp and the log phase culture was co-infected with VCS-M13 helper phage and grown in 2xTY-Amp-Kan overnight at 30 °C. The rescued phage was precipitated with polyethylene glycol and used for the next round of selection. The library was subjected to three rounds of selection and resulted in producing sets of polyclonal phage for each antigen, e.g., anti-C1q fraction. Serial dilutions of all polyclonal phage were plated on TYE-Amp for titer determination.

### 4.6. Selection of Monoclonal scFv

The plates with serial dilutions of the polyclonal phage from the three rounds were used for selecting random monoclonal scFv phage for each of the six anti-C1q fractions. The selection was on the basis of colony morphology. The selected clones were inoculated into 100 µL/well 2xTY-Amp in 96-well plates (Cell Well, Corning, Glendale, AZ, USA) and grown with shaking (190 revs/min) overnight at 37 ℃. The overnight cultures were transferred into a new 96-well plate by inoculation of 2 µL in 200 µL fresh 2xTY-Amp and cultivated for one hour. Then, 50 µL of 2xTY-Amp containing 10^10^ PFU of VCS-Ml3 was added to each well and the plate was incubated at 37 °C for 30 min without shaking. The cells were spun down by centrifugation at 3000 rev/min at 4 °C, resuspended in 200 µL/well 2xTY-Amp-Kan and grown overnight at 30 °C, 120 rev/min shaking. The supernatant containing monoclonal phage was tested for binding of antigen by immunoblot. Phage that displayed scFv were incubated with antigen-coated nitrocellulose membranes. Bound phage was detected by anti-M13-HRP (Sigma-Aldrich, Darmstadt, Germany) and DAB (diaminobenzidine, Sigma-Aldrich, St. Louis, MO, USA).

### 4.7. Induction of Soluble scFv and Affinity Purification

The soluble scFv was induced by two alternative induction protocols [49]. Briefly, for IPTG induction, *E. coli* HB2151/A1 cultures (OD_600nm_ of 0.8) were induced with 0.5 mM IPTG (Fisher Bioreagents, UK) for five hours at 25 °C. For autoinduction, overgrown *E. coli* HB2151/A1 daily cultures were inoculated 1:100 in ZYP-5052 autoinduction medium and grown for 16 h at 25 °C. The induced cells were lysed in 1/20th of the culture volume in ice-cold 100 mM Tris-HCl, pH.8.0, containing 20% sucrose (Fisher Scientific, Loughborough, UK) and 1 mM EDTA (Fisher Bioreagents, Fair Lawn, NJ, USA) for one hour on ice and then centrifuged at 9000 rpm for 45 min at 4 °C. The yielded supernatant (SN1), containing soluble scFv, was kept on ice and later pooled with SN2, obtained after the subsequent lysis of the cell pellet in the same volume of ice-cold 5 mM MgSO_4_ for 15 min on ice and then centrifuged at 9000 rpm for 45 min at 4 °C. The pooled supernatants were dialysed against phosphate buffer, pH 8.0 containing 10 mM Imidazole (Acros Organics, Fair Lawn, NJ, USA) and applied on HIS-Select^®^ Ni-affinity gel column (Sigma-Aldrich, Saint Louis, MO, USA) at a flow rate of 0.5 mL/min. The scFv antibodies were eluted with phosphate buffer, pH 8.0 containing 250 mM Imidazole. The eluted protein samples were dialysed against PBS, pH 7.2.

### 4.8. Enzyme-Linked Immunosorbent Assays

The inhibitory ELISA analyses were performed with four test-antigens—the native C1q (Merck Millipore Calbiochem™ Calbiochem, Darmstadt, Germany) and the recombinant analogues of ghA, ghB and ghC. The source of autoantibodies, designated IgG_LN_, was affinity purified IgG from pooled LN sera. Microtiter 96-flat bottom plates were coated with C1q or ghA, or ghB or ghC (1 µg/well in SC) and blocked with 1% BSA (Fisher Scientific, UK). Increasing amounts of the analysed clone scFv (2.5, 5 and 10 µg/well) was pre-incubated with biotinylated IgG_LN_ in PBS/0.75 for 30 min at room temperature and then added to the antigen-coated plates for an overnight incubation at 4 °C. Bound autoantibodies were detected by ExtrAvidin-AP (Sigma-Aldrich, St. Louis, MO, USA) and para-nitrophenylphosphate (pNPP, Life Technologies, Inc. Gaithersburg, MD, USA). The absorbance was read at 405 nm with a microplate reader (DR-200B, Hiwell Diatek Instruments, Wuxi, China). All samples were analysed in triplicate.

### 4.9. Spectroscopy Study of the Binding of Native C1q and ghA, ghB and ghC (Recombinant Fragments)

Emission spectra were registered by a Shimadzu spectrofluorophotometer RF 5000 (Shimadzu, Tokyo, Japan). The fluorescence spectroscopy experiments were performed in a 1 cm quartz cell (Hellma Analytics, Müllheim, Germany). C1q, ghA, ghB and ghC were studied using fluorescent dye ANS, excited at 390 nm with a slit width of 1.0 nm. Total fluorescence was calculated after normalizing the spectra and correcting for dilution. For the normalization of the inner filter and self-absorption effects, the excitation was always carried out less than 0.05 OD. Studies were done at 22 °C. The binding of C1q and ghA, ghB and ghC with increasing concentrations (0.03–0.3 µM) were added into the cuvette, containing LN autoantibodies complexed with scFv A1.

### 4.10. Sequence Analysis of Selected Clones

ScFv A1 coding sequence in pHEN2 vector was amplified with in situ polymerase chain reaction (PCR) of single colony *E. coli* HB2151, primers LMB3 (5′-CAGGAAACAGCTATGAC-3′) and fd-SEQ1 (5′-GAATTTTCTGTATGAGG-3′), according to amplification conditions of Liners et al. [50]. The PCR product was purified (GE Healthcare illustra™ GFX™ PCR DNA and Gel Band Purification Kit) and further sent for bidirectional Sanger sequencing (Macrogen, Amsterdam, the Netherlands). The derived sequence was manually aligned, cured from vector parts and deposited in GenBank under accession number KX981596.

### 4.11. scFv A1 Homology Model Generation

The MOE2016 software [51] was used for the homology modeling. Crystallized structures used for our homology modeling were derived from the Protein Data Bank (PDB). We performed alignment of the target sequence to the selected parts of best scored XRD structures of antibodies according to the modified version of the Needleman and Wunsch algorithm [52] with sequence alignment imposed on amino acid BLOSUM 62 substitution matrix, with algorithm penalty for gap opening -12 and gap extension -2 and evaluated with E-value accepted 1.e -12. The first five hits (PDB ID: 1YNL, 1A5F, 1ETZ, 1DEE, 1AD9) showing statistically significant similarity, the most similar one having E ~ 2.e -42, were chosen for further structure modeling of our sequence.

The homology model of scFv A1 was carried out by the Boltzmann-weighted randomized modeling procedure using coulomb and generalized Born implicit solvent interaction energies, using MMFF94 force field, as was implemented in MOE software suite (MOE2016).

The obtained 3D structure of scFv A1 sequence by homology modeling was further relaxed by molecular dynamics simulations with GROMACS software [53]. The structure of scFv A1 was neutralized and placed in periodic boundary conditions with explicit water molecules (TIP3P) with added Cl^−^ and Na^+^ ions up to physiological concentration of 0.154 M/L. The simulation was performed with CHARMM27 force field and the final production run was done on an NPT ansembl, 300K, 1atm, Berendsen thermostat and Parrinello-Rahman pressostat. The last 100 ns of equilibrated in physiological condition structure was used for further analysis. The selected trajectory part was used for cluster analysis and the frame closest to the center of the largest cluster was used further.

### 4.12. C1q Modeling

PDB structure was checked and several structural improperties were corrected. Other non-protein species were removed. The missing amino acids were placed according to the electron density map of the structures, despite the low electron densities there, with the COOT 0.8.8 software [54]. New protons were added in order to achieve right protonation state at 7.0 pH as the experimental study was carried out at physiological conditions. Further molecular dynamics simulation and analysis were done as in the case of the scFv A1 model.

### 4.13. Comparative Analysis of scFv A1 Homology Model and ghC1q Chains

Sequence comparison of modelled globular head domains of human C1q and H-chain of generated homology model of scFv A1 was performed, using secondary structure information of those sequences having associated atomic coordinates. The target sequence was sequence-to-group aligned to each one of ghC1q chains. Manual alignment was used for refining comparison of CDR frames of scFv A1 H-chain and loops in A-, B- and C- ghC1q.

## 5. Conclusions

We generated an anti-idiotypic scFv, clone A1, which inhibited approximately 50% the recognition of human C1q and its globular fragments ghA, ghB and ghC by the LN autoantibodies. The CDRs of scFv A1 showed structural similarity to the apical region of gC1q comprising parts of all globular fragments thus localizing a globular autoepitope of C1q of a conformational type. 

The V_H_ CDR2 of A1 mimics the ghA sequence GSEAD suggested as a cross-epitope between anti-DNA and anti-C1q antibodies.

Other potential structural analogs to C1q autoepitopes are the inhibitory monoclonal scFv F6, F9 and A12.

## Figures and Tables

**Figure 1 ijms-22-08288-f001:**
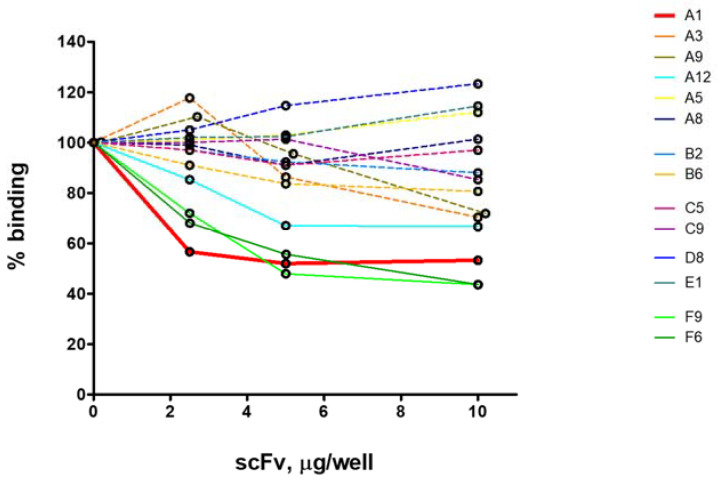
ELISA analysis of the inhibitory capacity of the anti-idiotypic monoclonal scFv antibodies on the recognition of immobilized C1q by IgG_LN_.

**Figure 2 ijms-22-08288-f002:**
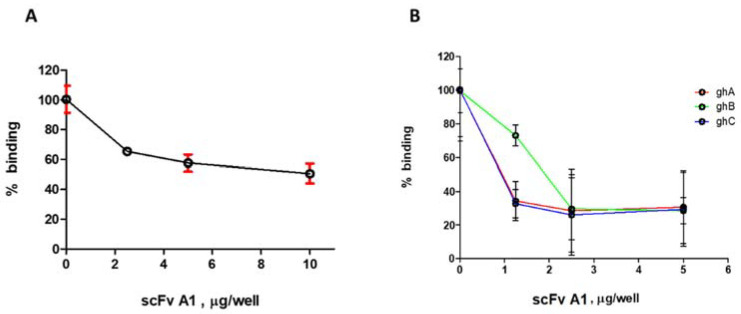
(**A**) ELISA analysis of the inhibitory capacity of scFv A1 on the recognition of immobilized C1q by IgG_LN_. Error bars (red) indicate mean ± SD. (**B**) ELISA analysis of the inhibitory capacity of scFv A1 on the recognition of immobilized ghA, ghB and ghC by IgG_LN_. Error bars (black) indicate mean ± SD.

**Figure 3 ijms-22-08288-f003:**
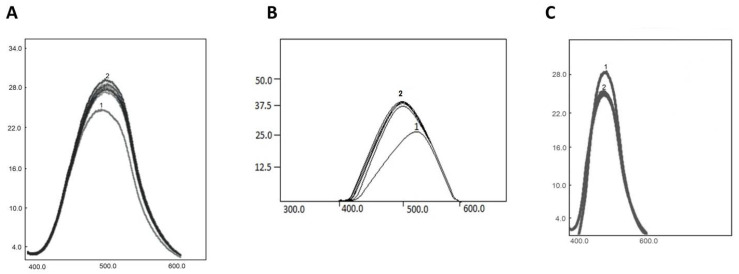
(**A**) Fluorescence spectra, showing ANS (1, lowest spectrum), LN autoantibodies complexed with scFv A1 (2, upper spectra) and C1q in increasing concentrations (0.03 µM–0.3 µM, overlapping the spectrum of the complex autoantibodies and scFv A1). (**B**) Fluorescence spectra, showing ANS (1, lowest spectrum), LN autoantibodies complexed with scFvA1 (2, upper spectra) and ghA in increasing concentrations (0.03 µM–0.3 µM, overlapping the spectrum of the complexed autoantibodies with scFv A1). (**C**) Fluorescence spectra, showing ANS (1, upper spectrum), LN autoantibodies complexed with scFvA1 (2, lower spectra) and ghC/ghB in increasing concentrations (0.03 µM–0.3 µM, overlapping the spectrum of the complexed autoantibodies with scFv A1).

**Figure 4 ijms-22-08288-f004:**
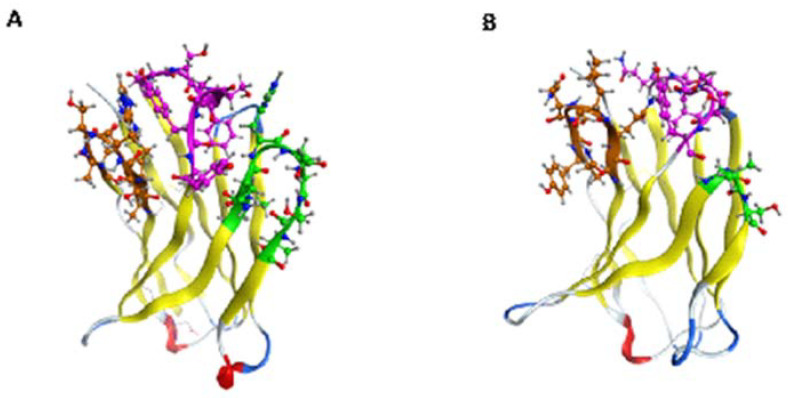
Representation of the generated model backbone of scFv A1 H-chain (**A**) and scFv A1 L-chain (**B**) in color coding (yellow beta structure, blue for loops). CDR1, 2 and 3 amino acids are depicted in magenta, green and brown, respectively. The revealed CDRs of H-chain are CDR1 (GGSFSGYY), CDR2 (INHSGST) and CDR3 (ARSHSAA). The revealed CDRs of L-chain are CDR1 (QGISSY), CDR2 (AAS) and CDR3 (QQLNSY).

**Figure 5 ijms-22-08288-f005:**
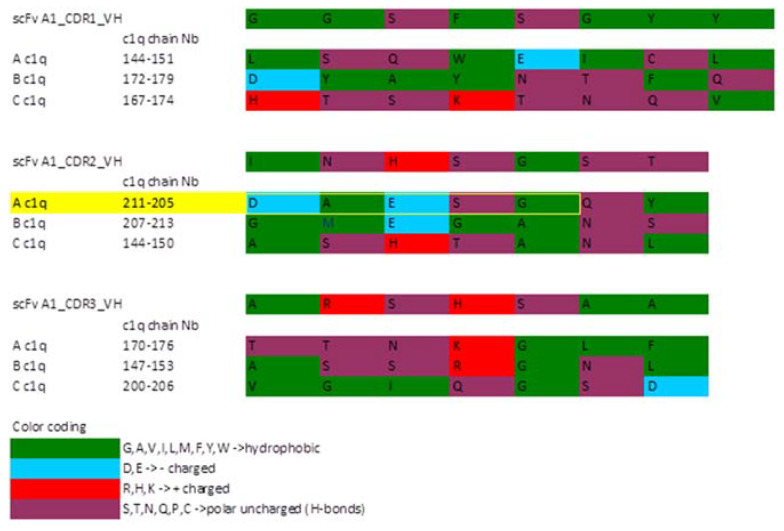
The optimal correlation of R polarity between C1q globular heads and CDR loops of H-chain scFv A1. The suggested DNA/C1q cross-reacting autoepitope GSEADV by Franchin et al. [24] from the ghA sequence is marked in yellow.

**Figure 6 ijms-22-08288-f006:**
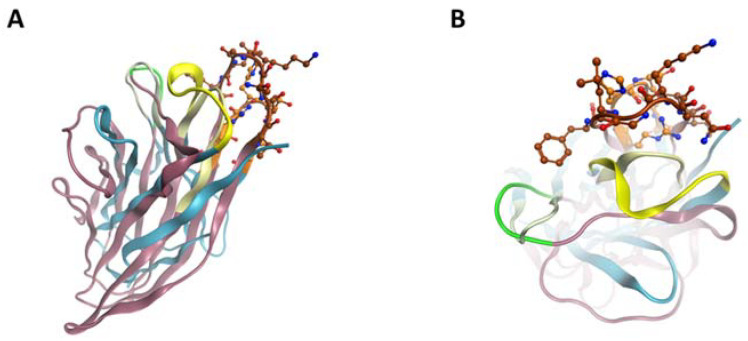
Side view (**A**) and top view (**B**) of V_H_ model (blue ribbons for frame regions) aligned on ghA of C1q (pink ribbons); CDR1, CDR2 and CDR3 of scFvA1 and ghA interface regions are presented in green, yellow and brown, respectively, with different color intensity in both chains.

**Figure 7 ijms-22-08288-f007:**
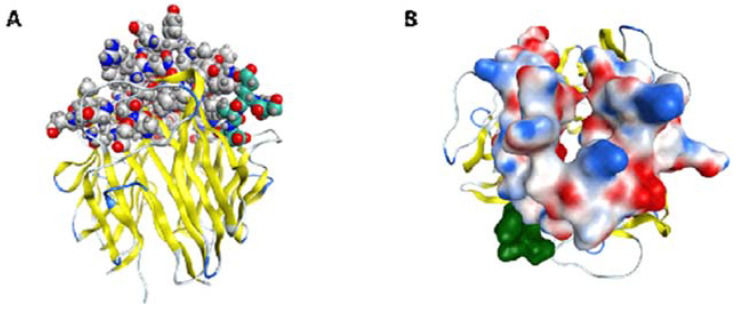
gC1q (yellow ribbons) parts, identified as structurally similar to scFvA1 V_H_ CDR1, 2 and 3 (grey, blue and red spheres); (**A**) Side view, where the cross-reactive DNA/C1q autoepitope ([24]; see Discussion) is clearly seen in light green colour. (**B**) Top view of the molecular surface with presented electrostatic potential, again in green is depicted the cross-reactive DNA/C1q autoepitope.

**Table 1 ijms-22-08288-t001:** Distribution of autoimmune sera into six groups according to the epitope specificities detected for them. Bolded are the common antigens for groups A, B and C (intact C1q) and the common test-antigen for groups D, E and F (ghC).

Sera Group/ Anti-C1q Fraction	Sera, Positive for Autoantibodies against:
A	intact **C1q**
B	intact **C1q** and CLR
C	intact **C1q** and CLR and ghC
D	**ghC**
E	CLR and/or ghA and/or ghB and **ghC**
F	ghA and/or ghB and **ghC**

**Table 2 ijms-22-08288-t002:** Immunoblotting of 68 rescued phage displayed scFv clones against anti-C1q fractions (A–F). Nitrocellulose membranes, coated with anti-C1q (A–F) and blocked with 0.5% Tween 20, were incubated with 10^12^ recombinant phage displayed scFv in TPBS from each clone. Bound phage was detected by anti-M13-HRP (1:2000 in TPBS) and DAB. The (+) and (−) refer to positive and negative signals, respectively. The (+/−) refers to a weak signal which was not considered positive.

Anti-C1q Fraction	1	2	3	4	5	6	7	8	9	10	11	12
A	+	−	+	−	+	−	−	+	+	−		+
B	−	+	−	−	−	+	−	−	+	−	−	−
C	+	−	−	−	+	−	−	−	+	−	−	−
D	−	+	−	−	−	−	+	+	−	−	−	
E	+	−	−	−	−	+/−	−	−	+/−	−	−	
F	−	−	+/−	−	−	+	−	−	+	−	−	

**Table 3 ijms-22-08288-t003:** Dot blotting of 21 monoclonal soluble scFv against anti-C1q fractions (A–F). Nitrocellulose membranes, coated with anti-C1q (A–F) and blocked with 0.5% Tween 20, were incubated with 2 mL of scFv-containing SN from each clone. Bound phage was detected by anti-C myc (9E10 mouse MoAb, 1:2000 in TPBS) and anti-mouse IgG-AP (1:2000 in TPBS), or alternatively by His-Probe-HRP (1:3 000 in TPBS). The anti-idiotypic clones are marked with (*). Human IgG was used as a control for isotypic epitopes, e.g., as a negative control. The (+) and (−) refer to positive and negative signals, respectively. The (+/−) refers to a weak signal which was not considered positive.

Antigen for Screening		C myc	His-Tag	Human IgGfrom Healthy Donors	Potentially Anti-Idiotypic scFv
Anti-C1q	A1	+/−	+	−	*
	A3	+/−	+	−	*
	A5	+/−	+	−	*
	A8	+/−	+	−	*
	A9	+/−	+	−	*
Anti-C1q and Anti-CLR	B2	+/−	+	−	*
	B6	+/−	+	−	*
	B9	+/−	+	+/−	
	A12	+/−	+	−	*
Anti-C1q and Anti-CLR and Anti-ghC	C1	+/−	+	+/−	
	C5	+/−	+	−	*
	C9	+/−	+	−	*
Anti-ghC	D2	+/−	+	+/−	
	D7	+/−	+	+/−	
	D8	+/−	+	−	*
Anti-CLR and/or Anti-ghA and/or Anti-ghB and Anti-ghC	E1	+/−	+	−	*
	E6	+/−	+	+/−	
	E9	+/−	+	+/−	
Anti-ghA and/or Anti-ghB and Anti-ghC	F3	+/−		+/−	
	F6	+/−	+	−	*
	F9	+/−	+	−	*

## Data Availability

The data that support the findings of this study are available from the corresponding author upon reasonable request.

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
