# Peer review of "Anti-Idiotype scFv Localizes an Autoepitope in the Globular Domain of C1q"

_ijms, 2021, doi:10.3390/ijms22158288_

Round 1

Reviewer 1 Report

This study tried to explore epitopes of complement C1q that are targeted by anti-C1q autoantibodies as occurring in patients with SLE. The exploration of these epitopes is of relevance as it helps understanding pathogenic mechanisms of the disease with a potential impact on treatment strategies. Autoantigenic sites of C1q are conformation dependent and possibly involve more than one of the three chains that form the molecule. Thus, these epitopes are difficult to detect by conventional epitope mapping. The authors chose the interesting approach to search for antiidiotypic antibodies targeting the antigen binding site of anti-C1q. Models of these anti-idiotypes were used to predict structures of C1q that are recognized by anti-C1q. As a primary result, the authors describe a structure on the globular heads that seems to be targeted by anti-C1q. I think, the aim of the study is of importance and the chosen strategy is highly interesting but I have doubts about the relevance of the findings as the methods have a high risk for producing misleading results and the relevance of autoantibodies against the globular heads of C1q is not well established (most anti-C1q bind to the collagen-like region). More specifically, I think the authors should address the following concerns:

  1. Abstract: Almost no introduction. Last 6 lines are hypotheses that are not supported by study data. Although these hypotheses are surely interesting to discuss they should not be part of the abstract.
  2. The study has a potential problem from start as elution of anti-C1q (using a Glycine buffer etc.) has a good chance to degenerate the antibodies. I understand that this step is difficult to avoid but the experiments lack an irrelevant control antibody having been exposed to the same sequence of buffers (most rigorous would be total IgG of an anti-C1q negative SLE patient). In line 109-111 it’s not getting clear whether this type of control was used (same for data shown in table 1-3). It is well possible that many of the scFv bind against the Fc regions.
  3. Figures 1 and 2: These experiments are critical for the whole study. Did the authors use a high-salt buffer and/or monomeric fractions of anti-C1q ? If not it could well be that the antibody preparations bind unspecifically to the globular head domains (due to low affinity interactions and/or due to aggregation of the purified antibodies). As a consequence, the detected autoantigenic site could be the antibody recognition domain of the globular heads. Again, data on control IgG should be shown.
  4. The last part of the results (starting at line 217) for me is rather a discussion and thus should be placed there.
  5. Discussion: I am missing a more critical discussion of the main findings, i.e. limitations of the approach (see concerns 2 and 3), possibility that the autoantigenic site is identical with the antibody binding domains of C1q (there are several papers that tried to identify this region and that could be incorporated. Is this region clearly different from the one described here ?), how do the authors explain the fact that they did not catch a structure on the collagen-like region (that should have had a higher chance).
  6. Minor:

Lines 259-264: C1q usually is not targeted by anti-C1q in the fluid phase but only when having bound to a specific structure. In my understanding, C1q therefore resembles a tissue-derived antigen.

Lines 269-272 (“This assumption …”): This statement requires a reference. Has this sequence of events clearly been shown (e.g. by an approach as published by Arbuckle et al., New Engl J Med 2003) ?

Lines 282-286: The link to EBV for me is not logical for the described antibodies. It would only fit if the scFv would resemble EBV not ‘anti-EBV’.

Last paragraph: Ref 34 described this type of interaction pretty well but is not discussed in this context. Do the anti-C1q fractions used in this study show crossreactivity with dsDNA ?

Author Response

Dear Reviewer,

Here are our notes to your comments:

1.Abstract: Almost no introduction. Last 6 lines are hypotheses that are not supported by study data. Although these hypotheses are surely interesting to discuss they should not be part of the abstract.

The first 2 lines of the Abstract are introduction. The next 1 line presents the research design. The following 6 lines summarize the main part of the results on which our hypothesis (next 3 lines) is based. And the last 2 lines describe additional results. On the whole we consider the Abstract well balanced and informative. We also regard the newly introduced hypothesis as important as the results leading to its formulation because the clinical relevance of anti-gC1q is at present being clarified.

2. The study has a potential problem from start as elution of anti-C1q (using a Glycine buffer etc.) has a good chance to degenerate the antibodies. I understand that this step is difficult to avoid but the experiments lack an irrelevant control antibody having been exposed to the same sequence of buffers (most rigorous would be total IgG of an anti-C1q negative SLE patient). In line 109-111 it’s not getting clear whether this type of control was used (same for data shown in table 1-3). It is well possible that many of the scFv bind against the Fc regions.

The term “degeneration” of the antibodies is biochemically undefined. Nevertheless, the IgG antibodies are routinely purified by affinity chromatography on Protein A/G and eluted by a Glycine buffer, pH 2.8. The exposure of immunoglobulins to this extreme pH is minimal as they are immediately neutralized in the process of collection and then equilibrated for a long period in physiological buffer. This technique is applied not only in research laboratories but also for the commercially available immunoglobulin preparations.

We do have a control IgG from pooled healthy donors’ sera which is clearly stated in the text (lines 110-111) and in the captions of Table 3.

The suggestion for using “total IgG of an anti-C1q negative SLE patient” is unsound as there aren’t anti-C1q negative patients. Even healthy humans have low titres of anti-C1q antibodies. We have used a negative control of pooled human IgG from healthy donors (the third column in Table 3). The scFv binding the Fc regions of the anti-C1q (e.g. the anti-isotypic scFv) were ruled out by using that same negative control (lines 110-111).

3. Figures 1 and 2: These experiments are critical for the whole study. Did the authors use a high-salt buffer and/or monomeric fractions of anti-C1q ? If not it could well be that the antibody preparations bind unspecifically to the globular head domains (due to low affinity interactions and/or due to aggregation of the purified antibodies). As a consequence, the detected autoantigenic site could be the antibody recognition domain of the globular heads. Again, data on control IgG should be shown.

We did use a high-salt buffer as it is explicitly stated in the text (lines 91-96) together with the explanation why the high-salt medium is needed.

IgG molecules with any antigenic specificity will bind the globular domains of C1q via the CH2 domains. The anti-C1q IgG molecules will bind C1q via the Fv domains. There is a distinct difference between the two interactions on the gC1q part. C1q binds the CH2 domain of IgG with a binding site extensively formed by ghB (Gaboriaud C et al., (2003) The crystal structure of the globular head of complement protein C1q provides a basis for its versatile recognition properties. J Biol Chem. 278(47), 46974-46982;  Kojouharova, M et al., (2004) Mutational analysis of the recombinant globular regions of human C1q A, B and C chains suggest an essential role for arginine and histidine residues in the C1q-IgG interaction. The Journal of Immunology 172 (7), 4351-4358), and not formed by the apex of the globular domain as it is demonstrated here by the structural analogue scFv A1.

4. The last part of the results (starting at line 217) for me is rather a discussion and thus should be placed there.

The suggestion is accepted, thank you.

5. Discussion: I am missing a more critical discussion of the main findings, i.e. limitations of the approach (see concerns 2 and 3), possibility that the autoantigenic site is identical with the antibody binding domains of C1q (there are several papers that tried to identify this region and that could be incorporated. Is this region clearly different from the one described here ?), how do the authors explain the fact that they did not catch a structure on the collagen-like region (that should have had a higher chance).

Yes, the localized autoepitope is different from the IgG-binding site of C1q as explained above.

We do expect to have produced a structural mimic of a CLR autoepitope by scFv A12 as is clearly stated in the text (lines 231-233; moved to Discussion in the revised manuscript) and A12 is now being examined.

6. Minor:

Lines 259-264: C1q usually is not targeted by anti-C1q in the fluid phase but only when having bound to a specific structure. In my understanding, C1q therefore resembles a tissue-derived antigen.

The term “tissue-derived antigen” refers to cytosolic antigens. That is why the necrotic cells are considered a source of autoantigens, which are released from the cytosol, and fuel further autoimmune response. C1q is a serum protein.

Lines 269-272 (“This assumption …”): This statement requires a reference. Has this sequence of events clearly been shown (e.g. by an approach as published by Arbuckle et al., New Engl J Med 2003) ?

Two references are inserted – 33 and 34.

Lines 282-286: The link to EBV for me is not logical for the described antibodies. It would only fit if the scFv would resemble EBV not ‘anti-EBV’.

The link to EBV would fit not only if the scFv (e.g.gC1q) would resemble EBV. It would also fit if C1q via gC1q binds directly to EBV. And C1q is known to bind directly viral proteins (gp40 of HIV for example). So far there is no experimental data to support the suggestion that C1q binds directly EBV but our finding is a clue for future investigations.

Last paragraph: Ref 34 described this type of interaction pretty well but is not discussed in this context. Do the anti-C1q fractions used in this study show crossreactivity with dsDNA ?

We do not report here experimental data showing that anti-C1q cross-react with dsDNA. What we do report is that scFvA1 contains a structural mimic of the previously suggested cross-epitope by Franchin et al. That is why these lines are part of our hypothesis rather than a statement.

Sincerely,

IvankaTsacheva

Reviewer 2 Report

Todorova et al. isolated four scFv's that are anti-idiotypic antibodies against fractions of LN anti-C1q auto-antibodies as antigens using phage panning. Among the four, scFv A1 was the most potent inhibitors of holo-C1q protein as well as three globular head domains, which are ghA, ghB, and ghC. 3D homology modeling of the scFv A1 and globular domain of C1q (gC1q) indicates structural similarity between the CDR regions of the scFv A1 and the apical region of gC1q. The amino acids sequence analysis also revealed that the auto-epitopes of C1q may be a conformational one because they are sparsely present on the gC1q. Furthermore, the authors proposed that a part of HCDR2 sequence of scFv A1, 207GSEAD211 as a cross-epitope of both C1q and dsDNA, suggesting that the anti-C1q auto-antibodies may act as a initial trigger of autoimmunity.

This study is very interesting in that the authors suggests globular head domains of C1q as auto-epitope for the first time by engineering anti-idiotypic scFvs that binds to six different anti-C1q auto-antibody fractions of LN sera: I would recommend acceptance to the IJMS; however, please refer to the following comments or suggestions.

  1. The whole manuscript needs English editing. For example,
    i) line 55: licalized --> localized
    ii) E.coli --> E. coli
    iii) the x- axis legend of Figure 2B may be scFv A1, parallel to that to Fugure 2A because the analytes are scFv A1.
    iv) line 209 does not seem to be a complete sentence.
    v) line 242 broun --> brown
    vi) lines 243-244 may be in the Figure 7 legend or in the main text
    iii) line 253 this data -->< these data
  2. Line 101: please reference for the Griffin.1 library or explain briefly
  3. Lines 184-188 may be included in the Figure 4.
  4. Figure 6 legend: please include what the pink and blue color depicts, which may be scFv A1 and gC1q, respectively.
  5. Figure 7A legend: please include how gC1q and scFv A1 are depicted in the structure (the yellow may be gC1q).

Author Response

Dear Reviewer,

Here are our notes to your comments:

1.The whole manuscript needs English editing. For example,
i) line 55: licalized --> localized

Corrected, thank you

ii) E.coli -->  coli

Corrected, thank you

iii) the x- axis legend of Figure 2B may be scFv A1, parallel to that to Fugure 2A because the analytes are scFv A1.

Corrected, thank you

iv) line 209 does not seem to be a complete sentence.

Corrected, thank you

v) line 242 broun --> brown

Corrected, thank you

vi) lines 243-244 may be in the Figure 7 legend or in the main text

Corrected, thank you

iii) line 253 this data -->< these data

Corrected, thank you

2. Line 101: please reference for the Griffin.1 library or explain briefly

Reference is included

3. Lines 184-188 may be included in the Figure 4.

The suggested lines are included in Figure 4

4. Figure 6 legend: please include what the pink and blue color depicts, which may be scFv A1 and gC1q, respectively.

The legend of Figure 6 is revised and colour coding is explained

5. Figure 7A legend: please include how gC1q and scFv A1 are depicted in the structure (the yellow may be gC1q).

The legend of Figure 7 is revised and colour coding is explained

Sincerely,

Ivanka Tsacheva

Reviewer 3 Report

The authors investigated the autoantigenicity of C1q in autoimmunity. They fractionated anti-C1q antibodies from patients with lupus nephritis, picked up scFv clones, and analyzed the binding affinity of these scFv to the different domains of C1q. And, using in silico analyses, they found the similarity of scFv A1 and gC1q, suggesting a cross-epitope between anti-DNA and anti-C1q antibodies. Finally, they hypothesized that the cross-reacting anti-dsDNA could be the antibodies dragging C1q initially into the context of autoimmune response by turning it into autoantigen.

The study is well conducted, and the manuscript is well written.

Their hypothesis is interesting.

Concerns

#1. The meaning of ± in Tables 2 & 3 should be clearly described.

#2. To strengthen their hypothesis, supporting clinical evidence should be discussed more deeply. They just described, “This assumption agrees with the observations that chronologically anti-Ciq autoantibodies appear after anti-dsDNA autoantibodies” without referring to any literature. Any related clinical studies should be added, e.g., chronological expression, if any, or the association of the expression of these antibodies.

Author Response

Dear Reviewer,

Here are our notes to your comments:

Concerns

#1. The meaning of ± in Tables 2 & 3 should be clearly described.

            A description is added to the captions of Tables 2 & 3 

#2. To strengthen their hypothesis, supporting clinical evidence should be discussed more deeply. They just described, “This assumption agrees with the observations that chronologically anti-Ciq autoantibodies appear after anti-dsDNA autoantibodies” without referring to any literature. Any related clinical studies should be added, e.g., chronological expression, if any, or the association of the expression of these antibodies.

            Two references reporting clinical data are included

Sincerely,

Ivanka Tsacheva

Round 2

Reviewer 1 Report

The abstract has not been changed. According to the authors’ own judgement it follows the unusual structure of … results-hypothesis-results-end.

The sentence ‘This assumption agrees with the observations that chronologically anti-C1q autoantibodies appear after anti-dsDNA autoantibodies’ now is referenced with two studies (33=Arbuckle et al., 34=Eriksson et al.) but I cannot find data on anti-C1q in them. Is there supplementary data that I missed ?

The statement in the response that ‘Even healthy humans have low titres of anti-C1q antibodies’ might be true but is not a generally accepted view. Is there published data on anti-C1q supporting this strong statement ?

The statement that ‘C1q is a serum protein’ in the response does not fully mirror the situation of C1q as an autoantigen, as C1q in serum is mostly in complex with C1r/s (forming C1), and C1 as well as free C1q are weak targets of anti-C1q, while bound C1q (maybe tissue bound) is. The term ‘tissue’ is not identical with ‘cytosolic’ as proposed.

Author Response

Dear Reviewer,

Here are our notes to your last comments:

1. The abstract has not been changed. According to the authors’ own judgement it follows the unusual structure of … results-hypothesis-results-end.

It isn’t unusual for an abstract to contain a hypothesis logically following the presented results. But as we do not report here direct experimental evidence for the cross-reaction of the two types of autoantibodies we decided to omit the sentence describing the hypothesis in the abstract.

2. The sentence ‘This assumption agrees with the observations that chronologically anti-C1q autoantibodies appear after anti-dsDNA autoantibodies’ now is referenced with two studies (33=Arbuckle et al., 34=Eriksson et al.) but I cannot find data on anti-C1q in them. Is there supplementary data that I missed ?

Exactly, anti-C1q antibodies are not among the first specificities of autoantibodies that are detected before the onset of the clinical manifestation of lupus. Anti-dsDNA antibodies are though, hence our assumption for the chronology of appearance of anti-DNA and anti-C1q.

3. The statement in the response that ‘Even healthy humans have low titres of anti-C1q antibodies’ might be true but is not a generally accepted view. Is there published data on anti-C1q supporting this strong statement ?

Yes, there are published data. One example is in Figure 1 of the article: Tsacheva I., Radanova M., Todorova N., Argirova T., and Kishore U.(2007). Detection of autoantibodies against the globular domain of human C1q in the sera of systemic lupus erythematosus patients. Molecular Immunology, (doi:10.1016/j.molimm.2006.09.009), 44(8):2157-61.

Pooled serum from healthy donors is used in every study measuring autoantibody titers of any specificity as the control value registering the cutoff from the values of the patients.

4. The statement that ‘C1q is a serum protein’ in the response does not fully mirror the situation of C1q as an autoantigen, as C1q in serum is mostly in complex with C1r/s (forming C1), and C1 as well as free C1q are weak targets of anti-C1q, while bound C1q (maybe tissue bound) is. The term ‘tissue’ is not identical with ‘cytosolic’ as proposed.

The term “tissue” is not identical to “cytosolic”. But the term “tissue-derived” is synonymous to “cytosolic” when autoantigens are meant by the authors of the reference in question (32. Rosen A, Casciola-Rosen L (2016) Autoantigens as partners in initiation and propagation of autoimmune rheumatic diseases. Ann. Rev Immunol. 34, 395-420.). 

Rosen and co-authors discuss tissue-derived molecules released from cellular or subcellular compartments as autoantigens (and C1q is not among them) and put them in the center of their resonance hypothesis which we discussed in contrast to extracellular autoantigens like C1q. The resonance hypothesis is a sound concept as the distinction intracellular/extracellular is the basis for the induction of immune system tolerance. All sequestered molecules (e.g. intracellular) are not considered “self” in contrast to extracellular molecules which are treated with immune tolerance. Therefore, C1q being an extracellular protein is non-immunogenic. This raises the question why anti-C1q antibodies appear. The answer could not be – “Because C1q is bound (maybe tissue bound)”. This is the physiological function of the protein as part of the Classical Complement. If every act of immobilization would turn C1q into autoantigen then the Complement System could not function in a physiological mode.

It is true that C1q autoantigenicity is a complex issue. For this part of our research we decided to use the backwards approach – to identify the structural features of the autoepitopes that would lead us to the elusive immunogen(s). So far our clues lead us to the suggested cross-reactivity with anti-DNA, hence the hypothesis. Our on-going research is providing experimental evidence that further support this hypothesis. Other research group has suggested EBV to be the immunogen for the anti-CLR antibodies of C1q as discussed in our manuscript (36. Csorba K, Schirmbeck L A, Tuncer E, Ribi C, Roux-Lombard P, Chizzolini C, Huynh-Do U, Vanhecke D, Trendelenburg M (2019) Anti-C1q antibodies as occurring in systemic lupus erythematosus could be induced by an epstein-barr virus-derived antigenic site. Frontiers in immunology 10, 2619.).

Most of all our approach differs from most of the published studies with the fact that we included in the cohort active patients as well as non-active (which were meant as a control for the clinical activity of LN). We regarded the clinically non-active patients to be at the closest to the state of initiation of the next active phase thus providing clues for the circumstances contributing to C1q autoantigenicity. Anti-gC1q prevailed in those non-active patients (15. Stoyanova V, Tchorbadjieva M, Deliyska B, Vasilev V, Tsacheva I (2012) Biochemical analysis of the epitope specificities of anti-C1q autoantibodies accompanying human Lupus Nephrites reveals them as a dynamic population in the course of the disease. Imm.Lett 148, 69–76.), while anti-CLR antibodies prevailed in clinically active patients.  It was clear to us that the two fractions of anti-C1q had different contribution to the disease phases – initiation, progression, and self-sustenance.